## Research Article

eHealth literacy; online health information seeking; mental health; adolescence; COVID-19

**Corresponding author:**
Fong-Ching Chang;
Email: fongchingchang@ntnu.edu.tw

# Adolescent pursuit of health information online during the COVID-19 pandemic: The roles played by eHealth literacy and psychological distress

Fong-Ching Chang[1] , Chingching Chang[2] and Chen-Chao Tao[3]

[1]Department of Health Promotion and Health Education, National Taiwan Normal University, Taipei, Taiwan; [2]Research Center for Humanities and Social Sciences, Academia Sinica, Taipei, Taiwan and [3]Department of Communication and Technology, National Yang Ming Chiao Tung University, Hsinchu, Taiwan

## Abstract

COVID-19 has led to an increase in mental health problems for adolescents. In this study, we examined the factors related to the eHealth literacy of adolescents and how that impacted their pursuit of health information and mental health information online during the COVID-19 pandemic. We analyzed data from the 2020 Taiwan Communication Survey, which involved a total of 1,250 national representative adolescents who completed an online questionnaire. The results showed that two-thirds of adolescents reported searching for health information online, and about half of them reported searching for mental health information online during the COVID-19 pandemic. Multivariate analysis results indicated that adolescents who spent more time learning online, had higher levels of bonding social capital and self-determination, and had higher levels of parental active internet mediation were more likely to have higher levels of eHealth literacy. In addition, multivariate analysis results showed that adolescents who had higher levels of eHealth literacy and had higher depression and anxiety were more likely to seek health information and mental health information online. In conclusion, the levels of eHealth literacy and psychological distress of adolescents played a crucial role in their pursuit of health information and mental health information online during the COVID-19 pandemic.

## Impact statement

Adolescents are showing an increase in vulnerability to mental health illness, and studies have documented an increase in psychological distress among adolescents during the COVID-19 pandemic. Adolescents spent more time online, and their pursuit of mental health information online has facilitated their autonomy to control their help-seeking journey and allow them to connect with others with more privacy and less stigma. The abundance of misinformation online, however, particularly during the COVID-19 outbreak, has enhanced the need for eHealth literacy in seeking health information online, appraising health information, making healthy decisions and implementing protective behaviors. This study examined the role of factors such as self-determination, eHealth literacy and psychological distress in determining adolescents' pursuit of health / mental health information online during the COVID-19 pandemic.

## Introduction

Adolescents are showing an increase in vulnerability to mental health illness (Gunnell et al., 2018), and studies have documented an increase in psychological distress among adolescents during the COVID-19 pandemic (Nearchou et al., 2020). School closures, social distancing measures and isolation had negatively impacted the mental health of children and adolescents during the COVID-19 pandemic (Meherali et al., 2021). Adolescents tend to be reluctant to seek mental health services in person, but may seek mental health information online due to less stigma and more privacy (Pretorius et al., 2019). Studies have shown that the recent pursuit of mental health information online has been high among adolescents (Park and Kwon, 2018; Rideout et al., 2018), and that individuals with psychological distress and mental health problems were more likely to pursue health information online (Gallagher and Doherty, 2009; Rowlands et al., 2015; Pretorius et al., 2019). A review study found that for young people with a preference for self-reliance, the pursuit of mental health information online is either therapeutic or acts as a gateway to further help (Pretorius et al., 2019).

Children and adolescents spend more time online than adults, which is a global phenomenon. The 2020 EU Kids Online survey showed that in many countries the time that children report spending online almost doubled compared with the 2010 findings (Smahel et al., 2020). In particular, during the COVID-19 pandemic adolescents spent more time than ever on the

internet, which they used for socializing, entertainment and learning (Ofcom, 2020). The internet is the main source of information for adolescents, while a review study found that the primary purpose for adolescents' health-related use of the internet is the pursuit of health information (Park and Kwon, 2018). Studies showed that about half of adolescents have reported using the internet to search for health information (Jiménez-Pernett et al., 2010; Gazibara et al., 2020), but many of those adolescents also reported a lack of search skills to find reliable web pages and had difficulties in filtering overabundant content and determining the quality of information (Esmaeilzadeh et al., 2018; Patterson et al., 2019).

The COVID-19 pandemic has provoked a greater number of healthcare organizations to provide eHealth resources and to stress the importance of individual eHealth literacy for the use of eHealth services. The abundance of misinformation online, particularly during the COVID-19 outbreak, has enhanced the role of eHealth literacy in appraising online information, making health decisions and implementing protective behaviors (Brørs et al., 2020). eHealth literacy refers to an individual's ability to seek out, understand, appraise and apply electronic health information to solve health problems (Norman and Skinner, 2006). A study has shown that individuals with higher eHealth literacy were more likely to search for COVID-19 information online and adopt preventive behaviors (Guo et al., 2021). Prior studies also found a positive association between higher levels of eHealth literacy and the pursuit of health information online (Gazibara et al., 2020), healthy lifestyle behaviors in adolescents (Gürkan and Ayar, 2020; Eyimaya et al., 2021) and positive mental health effects (Chen et al., 2020). Most studies assessed eHealth literacy using self-reporting of perceived skills, while few studies have measured performance-based eHealth literacy through the testing of actual performance to determine eHealth literacy levels, such as the completion of computerized simulation tasks. Prior studies showed that perceived and performed eHealth literacy were either moderately (Neter and Brainin, 2017) or weakly correlated (van der Vaart et al., 2011). Some studies have shown a large discrepancy between perceived and performance-based eHealth literacy, which has highlighted the levels of poor critical self-awareness among adolescents (Maitz et al., 2020; McKinnon et al., 2020.

During the COVID-19 pandemic, parents and adolescents were living with increased stress (Cluver et al., 2020). Parents have reported increases in the difficulties associated with controlling their children's screen time and increased levels of concern related to the online risks to children (Ofcom, 2021). When adolescents spend more time using the internet, online risks and psychological distress increase (Deslandes and Coutinho, 2020; Guessoum et al., 2020). Parents play a crucial role in the use of the internet by their children. A review study found that the parent–child relationship, parental mediation practices and parents' own use of media influences children's media use, attitudes and effects (Coyne et al., 2017). A study found that parental eHealth literacy, active parental internet mediation and adolescent internet health information literacy all were related to the pursuit of health information online by adolescents (Chang et al., 2015).

Self-determination theory emphasizes that the satisfaction of basic psychological needs including competence, autonomy and relatedness were associated with development, health-behavior change and better mental health (Deci and Ryan, 2000). A prior study associated a higher level of self-determination with seeking online health information (Lee and Lin, 2016). Despite studies that have examined the relationships between eHealth literacy and the pursuit of health information online and related factors, most

research has focused on adults. By comparison, only a scant amount of research has explored adolescent pursuit of health information online and examined the influence of self-determination, eHealth literacy, psychological distress and parental mediation. For the present study, we analyzed data from the 2020 Taiwan Communication Survey conducted during the COVID-19 pandemic. Our aims included (1) an assessment of adolescents' pursuit of health information and health topics online during the COVID-19 pandemic; (2) an examination of the relationships between internet use, self-determination and parental internet mediation with adolescents' eHeatlh literacy; and (3) a further examination into how self-determination, eHealth literacy, psychological distress and parental internet mediation are associated with adolescents' pursuit of health information online (i.e., the pursuit of health/ mental health information online and the number of health topics searched online).

## Methods

### Participants

This study analyzed data from the 2020 Taiwan Communication Survey (second phase, fourth wave) (Chang, 2022). The cluster sampling method was used, while the six socioeconomic strata from Taiwan townships (Hou et al., 2008) were adopted as a sampling scheme to draw the sample schools. A total of 19 elementary schools, 20 middle schools and 23 high schools were randomly selected from the strata and invited to join the survey. The classes were randomly selected from the sample schools. Teachers gave students consent forms to take home to parents requesting consent for their children to participate in the survey. Students were assured that their information would be protected and anonymous. This study analyzed 1,250 national representative adolescents from 43 middle schools (n = 582) and high schools (n = 668). Adolescents completed an online self-administered questionnaire between November 2020 and January 2021, during the COVID-19 pandemic. Approval was obtained from the Institutional Review Board at Academia Sinica, Taiwan.

A self-administered questionnaire was developed based on previous studies (Norman and Skinner, 2006; Williams, 2006; Löwe et al., 2010; Nishimura and Suzuki, 2016; Rideout et al., 2018). A group of experts was invited to assess the content validity of the questionnaire. In addition, a pilot survey was conducted at one middle school and two high schools in order to assess the appropriateness of the survey questions and to evaluate the reliability of the data that the questionnaire would yield.

### Pursuit of health information online

Questions concerning the pursuit of health information online were adapted from a study found in the literature (Rideout et al., 2018). Participants were asked, "Have you used the internet to search the following health topics?" If participants answered that they did not use the internet to search for any health topics, then they were categorized as not seeking health information online. If participants answered that they use the internet to search for health topics (i.e., exercise, physical fitness, nutrition, sexual health, stress management, depression, anxiety, mental health and others), then they were categorized as having sought health information online. In addition, if participants answered that they used the internet to search for any mental health topics (i.e., sexual health, stress management, depression, anxiety and mental health), then they were categorized as having sought mental health information online.

### Internet use time

Internet use time was measured using six items. Participants were asked how much time (amount of hours and minutes) they spend on computers, tablets and/or smartphones for learning and recreation, respectively. Sample questions follow: "How long do you surf the internet using a computer (excluding the usage of smartphones and tablets; only including your internet-surfing screen time) for learning and working purposes every day?"; "Excluding your learning and working time online, how long do you surf the internet using a computer for entertainment and leisure purposes every day on average?" The sum of internet use time on a computer, a tablet, or a smartphone for learning and working purposes was considered internet learning time. On the other hand, the sum of time using the internet on a computer, a tablet, or a smartphone for entertainment and leisure purposes was considered internet entertainment and leisure time.

### eHealth literacy

eHealth Literacy (5 items) was measured by adapting a previously developed eHealth literacy scale (Norman and Skinner, 2006) to measure participants' perceived skills at finding, evaluating and applying electronic health information to health problems. Participants were asked about their opinions and experiences using the internet for health information. Sample statements follow: "I know how to find helpful health resources on the internet"; "I know how to use the internet to answer my health questions"; "I know how to use the health information I find on the internet to help me"; "I can tell high-quality from low-quality health resources on the internet"; "I feel confident in using information from the internet to make health decisions." The response options were graded on a 5-point Likert-type scale that ranged from strongly disagree (scoring 1) to strongly agree (scoring 5), with higher scores indicating a higher level of eHealth literacy. The Cronbach's alpha of eHealth literacy was 0.87.

### Parental internet mediation

Parental internet mediation (six items) was measured by adapting scales from prior studies (Nathanson, 1999; Valkenburg et al., 1999). Parental internet mediation included parental restrictive internet mediation (three items) and parental active internet mediation (three items). Participants were asked about their experience regarding their parents' internet mediation. Parental restrictive internet mediation statements follow: "My parents don't allow me to visit certain websites"; "My parents set rules regarding when I can use the internet and when I cannot"; "My parents set a limit on how long I use the internet." Active parental internet mediation statements follow: "My parents encourage me to use the internet"; "My parents discuss internet use experiences with me"; "My parents discuss online stories and events with me." The response options included never (scoring 1), seldom (scoring 2), sometimes (scoring 3), often (scoring 4), with higher scores indicating a higher level of parental restrictive/active mediation. The Cronbach's alpha scores for parental restrictive internet mediation and for active parental internet mediation were 0.76 and 0.68, respectively.

### Bonding social capital

Bonding social capital (one item) was measured by adapting an approach from a social capital study (Williams, 2006) to measure participants' perceived emotional support and access to resources from strong-tie networks. Participants were asked to complete the following statement: "There are several people I trust to help solve my problems." The response options were graded on a 5-point Likert-type scale that ranged from strongly disagree (scoring 1) to strongly agree (scoring 5), with higher scores indicating a higher level of bonding social capital.

### Self-determination

Self-determination (three items) was measured by factors that included competence, autonomy and relatedness, and was adapted from a study found in the literature (Nishimura and Suzuki, 2016). Participants were asked their opinions and experiences regarding the following statements: "I feel confident that I can do things well"; "I feel I have been doing what really interests me"; "I feel close and connected with other people who are important to me (e.g. family, friends)." The response options were graded on a 5-point Likert-type scale that ranged from strongly disagree (scoring 1) to strongly agree (scoring 5), with higher scores indicating higher levels of self-determination (competence, autonomy and relatedness).

### Depression and anxiety

Depression and anxiety (four items) were measured using a scale developed in a previous study (Löwe et al., 2010). Participants were asked "Over the last week, how often have you been bothered by the following problems? (1) Little interest or pleasure in doing things; (2) Feeling down, depressed or hopeless; (3) Feeling nervous, anxious or on edge; (4) Not being able to stop or control worrying." The response options ranged from 0 (scoring 0) to 7 days (scoring 7), with higher scores indicating a higher level of depression and anxiety. The Cronbach's alpha of depression and anxiety was 0.90.

### Physical and emotional exhaustion

To gauge physical and emotional exhaustion, two items were adopted from the Copenhagen Burnout Inventory (Kristensen et al., 2005). The questions were "How often are you physically exhausted?" and "How often are you emotionally exhausted (such as feeling helpless or frustrated)?" The response options included never (scoring 1), seldom (scoring 2), sometimes (scoring 3) and often (scoring 4), with higher scores indicating a higher level of physical and emotional exhaustion.

### Characteristics of the adolescents

The characteristics of the adolescents who participated in this study included gender (male or female), age, school type (middle school or high school) and academic performance (very good, good, average, poor, very poor).

### Statistical analysis

SAS was used to perform the statistical analysis. A series of t-tests were conducted to compare adolescents' eHealth literacy, internet use time and related factors by gender and by school type. Chi-square tests were conducted to compare adolescents' pursuit of health information and mental health online. In addition, multiple regression was conducted to examine the factors related to adolescents' eHealth literacy and the number of health topics searched online. Multiple logistic regression was conducted to examine factors related to adolescents' pursuit of health information online and the factors associated with the pursuit of mental health information online. A 95% confidence interval (95% C.I.) and *p* value was presented. The outcome variable was the pursuit of health information online including the pursuit of health information online, pursuit of mental health information online and number of health topics searched online. The independent variables included internet use time, eHealth literacy, parental internet

mediation, bonding social capital, self-determination, depression and anxiety, physical and emotional exhaustion. The covariate variables were adolescents' characteristics including gender, age and academic performance.

## Results

### Internet use by adolescents

Of the 1,250 participant students, 655 were boys (52.4%) and 595 were girls (47.6%). The age range of participant students was 12–18 years, while the mean age of middle school students and high school students was 12.9 years, and 16.1 years, respectively. Overall, adolescents spent 29.8 h per week using the internet

for recreation and 12.5 h per week using the internet for learning. In addition, adolescents mainly accessed the internet via smartphones (22.2 h/week for recreation and 8.3 h/week for learning). By gender, boys spent more time using computers to go online than girls. By school type, high school students spent more time using computers and smartphones to go online than middle school students (Table 1).

### eHealth literacy of adolescents and related factors

The eHealth literacy of the adolescent participants was above average (Mean = 3.69) (Table 1). The participants had higher scores for the searching of health information than for appraising and

**Table 1.** Adolescent's internet use, eHealth literacy and health status

|  | Total Mean (SD) | Girl Mean (SD) | Boy Mean (SD) | t-test *p* value | Middle school Mean (SD) | High school Mean (SD) | t-test *p* value |
|---|---|---|---|---|---|---|---|
| Age | 14.60 | 14.51 | 14.68 | 0.0987 | 12.91 | 16.08 | <0.0001 |
|  | (1.80) | (1.81) | (1.79) |  | (0.84) | (0.97) |  |
| Internet learning time (hr/wk) | 12.47 | 12.37 | 12.57 | 0.8277 | 10.14 | 14.50 | <0.0001 |
|  | (15.98) | (15.93) | (16.03) |  | (14.87) | (16.63) |  |
| Computer(hr/wk) | 3.36 | 2.67 | 3.99 | 0.0007 | 2.13 | 4.43 | <0.0001 |
|  | (6.97) | (5.97) | (7.72) |  | (5.57) | (7.84) |  |
| Tablet(hr/wk) | 0.79 | 0.63 | 0.93 | 0.0936 | 1.04 | 0.57 | 0.0139 |
|  | (3.27) | (2.83) | (3.61) |  | (3.90) | (2.57) |  |
| Smartphone(hr/wk) | 8.32 | 9.07 | 7.64 | 0.0393 | 6.97 | 9.50 | 0.0002 |
|  | (12.15) | (12.93) | (11.36) |  | (11.31) | (12.72) |  |
| Internet recreation time (hr/wk) | 29.84 | 27.65 | 31.84 | 0.0075 | 24.33 | 34.65 | <0.0001 |
|  | (27.83) | (25.77) | (29.46) |  | (27.15) | (27.54) |  |
| Computer(hr/wk) | 5.98 | 3.41 | 8.31 | <0.000 | 3.80 | 7.88 | <0.0001 |
|  | (12.94) | (8.55) | (15.56) | 1 | (10.38) | (14.57) |  |
| Tablet(hr/wk) | 1.72 | 1.50 | 1.92 | 0.2913 | 2.18 | 1.32 | 0.0335 |
|  | (6.98) | (6.78) | (7.16) |  | (8.21) | (5.67) |  |
| Smartphone(hr/wk) | 22.15 | 22.74 | 21.61 | 0.3355 | 18.35 | 25.46 | <0.0001 |
|  | (20.79) | (20.91) | (20.69) |  | (20.78) | (20.25) |  |
| Health topic searches | 2.21 | 2.24 | 2.19 | 0.6534 | 1.76 | 2.60 | <0.0001 |
|  | (2.06) | (2.08) | (2.04) |  | (1.84) | (2.15) |  |
| EHealth literacy | 3.69 | 3.64 | 3.73 | 0.0118 | 3.68 | 3.70 | 0.6957 |
|  | (0.65) | (0.60) | (0.69) |  | (0.71) | (0.59) |  |
| Bonding social capital | 3.88 | 3.90 | 3.87 | 0.4889 | 3.84 | 3.91 | 0.1787 |
|  | (0.90) | (0.90) | (0.89) |  | (0.98) | (0.82) |  |
| Competence | 3.47 | 3.35 | 3.58 | <0.0001 | 3.47 | 3.48 | 0.7647 |
|  | (0.88) | (0.84) | (0.90) |  | (0.86) | (0.89) |  |
| Autonomy | 3.53 | 3.42 | 3.62 | 0.0003 | 3.59 | 3.47 | 0.0189 |
|  | (0.96) | (0.94) | (0.97) |  | (0.94) | (0.97) |  |
| Relatedness | 4.00 | 4.04 | 3.96 | 0.0992 | 3.94 | 4.04 | 0.0393 |
|  | (0.87) | (0.84) | (0.90) |  | (0.90) | (0.84) |  |
| Depression and anxiety | 1.56 | 1.79 | 1.36 | <0.0001 | 1.30 | 1.80 | <0.0001 |
|  | (1.66) | (1.73) | (1.58) |  | (1.56) | (1.72) |  |

*(Continued)*

**Table 1.** (*Continued*)

| | Total Mean (SD) | Girl Mean (SD) | Boy Mean (SD) | t-test *p* value | Middle school Mean (SD) | High school Mean (SD) | t-test *p* value |
|---|---|---|---|---|---|---|---|
| Physical exhaustion | 2.88 | 2.96 | 2.81 | 0.0010 | 2.67 | 3.06 | <0.0001 |
| | (0.77) | (0.75) | (0.78) | | (0.77) | (0.72) | |
| Emotional exhaustion | 2.89 | 3.02 | 2.78 | <0.000 | 2.69 | 3.07 | <0.0001 |
| | (0.82) | (0.79) | (0.82) | 1 | (0.83) | (0.76) | |
| Parental restrictive internet mediation | 2.13 | 2.10 | 2.16 | 0.2171 | 2.50 | 1.83 | <0.0001 |
| | (0.85) | (0.82) | (0.87) | | (0.86) | (0.71) | |
| Parental active internet mediation | 2.28 | 2.37 | 2.18 | <0.000 | 2.28 | 2.27 | 0.8638 |
| | (0.76) | (0.76) | (0.76) | 1 | (0.76) | (0.77) | |

*Note*: Boy n = 655 girl n = 595, middle school n = 582 high school n = 668.

applying the health information they found. By gender, boys had slightly higher levels of eHealth literacy (Mean = 3.73) than girls (Mean = 3.64) (Table 1).

Multiple regression results indicated that adolescents who were boys, who spent more time using the internet for learning, who had higher levels of bonding social capital, who higher self-determination (competence, relatedness), and who had higher levels of active parental internet mediation were more likely to have higher levels of eHealth literacy (Table 3).

### *Pursuit of health information online by adolescents and related factors*

The rates of adolescents' pursuit of health information online are listed in Table 2. Overall, 75.6% of adolescents searched health information online, while 47.7% of adolescents searched mental health information online during the COVID-19 pandemic. The percentage of girls who searched mental health topics online (50.8%) was higher than that for boys (44.9%). The percentage of high school students who searched health topics online (79.5%) was higher than that for middle school students (71.5%). Similarly, the percentage of high school students who searched mental health topics online (60.1%) was higher than that for middle school students (33.5%) (Table 2).

Multiple logistic regression results showed that adolescents who were boys, who had higher levels of eHealth literacy, who had higher levels of depression and anxiety, or who had higher levels of parental active internet mediation were more likely to seek health information online. In addition, adolescents who were high school students, who had higher levels of eHealth literacy, who had higher

**Table 2.** Adolescents' pursuit of health information online and health topics searched

| | Total n (%) | Girl n (%) | Boy n (%) | Chi-square *p* value | Middle school n (%) | High school n (%) | Chi-square *p* value |
|---|---|---|---|---|---|---|---|
| Pursuit of health information online | | | | 0.0684 | | | 0.0015 |
| No | 305 | 159 | 146 | | 166 | 139 | |
| | (24.4) | (26.7) | (22.3) | | (28.5) | (20.8) | |
| Yes | 945 | 436 | 509 | | 416 | 529 | |
| | (75.6) | (73.3) | (77.7) | | (71.5) | (79.2) | |
| Pursuit of mental health information online | | | | 0.0379 | | | <0.0001 |
| No | 654 | 293 | 361 | | 387 | 267 | |
| | (52.3) | (49.2) | (55.11) | | (66.5) | (40.0) | |
| Yes | 596 | 302 | 294 | | 195 | 401 | |
| | (47.7) | (50.8) | (44.9) | | (33.5) | (60.0) | |
| Health topics searched | | | | | | | |
| Exercise | 663 | 272 | 391 | | 315 | 348 | |
| | (53.0) | (45.7) | (59.7) | | (54.1) | (52.5) | |
| Physical fitness | 272 | 96 | 176 | | 138 | 134 | |
| | (21.8) | (16.1) | (26.9) | | (23.7) | (20.1) | |
| Nutrition | 352 | 163 | 189 | | 145 | 207 | |
| | (28.2) | (27.4) | (28.9) | | (24.9) | (31.0) | |

(*Continued*)

**Table 2.** (*Continued*)

|  | Total n (%) | Girl n (%) | Boy n (%) | Chi-square *p* value | Middle school n (%) | High school n (%) | Chi-square *p* value |
|---|---|---|---|---|---|---|---|
| Sexual health | 163 | 36 | 127 |  | 37 | 126 |  |
|  | (13.0) | (6.1) | (19.4) |  | (6.4) | (18.9) |  |
| Stress management | 388 | 210 | 178 |  | 110 | 278 |  |
|  | (31.0) | (35.3) | (27.2) |  | (18.9) | (41.6) |  |
| Depression | 288 | 182 | 106 |  | 83 | 205 |  |
|  | (23.0) | (30.6) | (16.2) |  | (14.3) | (30.7) |  |
| Anxiety | 240 | 147 | 93 |  | 68 | 172 |  |
|  | (19.2) | (24.7) | (14.2) |  | (11.7) | (25.8) |  |
| Mental health | 398 | 172 | 226 |  | 126 | 272 |  |
|  | (31.8) | (26.3) | (38.0) |  | (21.7) | (40.7) |  |

*Note*: Boy n = 655 girl n = 595, middle school n = 582 high school n = 668.

**Table 3.** Factors related to adolescents' eHealth literacy

|  | β | 95% C.I. | *p* value |
|---|---|---|---|
| Intercept | 2.06 | 1.65–2.48 | <0.0001 |
| Gender (boy = 1, girl = 0) | 0.10 | 0.03–0.17 | 0.0040 |
| Age | 0.00 | −0.02–0.02 | 0.9873 |
| Academic performance | 0.01 | −0.03–0.05 | 0.7134 |
| Internet learning time | 0.01 | 0.01–0.01 | 0.0178 |
| Internet recreation time | 0.01 | −0.01–0.01 | 0.3104 |
| Bonding social capital | 0.17 | 0.13–0.22 | <0.0001 |
| Competence | 0.10 | 0.06–0.15 | <0.0001 |
| Autonomy | 0.01 | −0.04–0.05 | 0.7887 |
| Relatedness | 0.08 | 0.03–0.12 | 0.0010 |
| Parental restrictive internet mediation | −0.01 | −0.05–0.03 | 0.6695 |
| Parental active internet mediation | 0.09 | 0.05–0.14 | <0.0001 |

*Note*: (1) N = 1,236. (2) Multiple regression was conducted.

levels of depression and anxiety, and who had higher levels of emotional exhaustion were more likely to seek mental health information online (Table 4).

### Health topics searched by adolescents and related factors

The health topics that adolescents searched included exercise (53.0%), mental health (31.8%), stress (31.0%), nutrition (28.2%), depression (23.0%), physical fitness (21.8%), anxiety (19.2%), and sexual health (13.0%) (Table 2). The average number of health topics adolescents searched was 2.2. High school students searched more health topics (mean = 2.60) than middle school students (1.76) (Table 1). Multiple regression results showed that adolescents who were high school students, who had higher levels of eHealth literacy, who had higher competence, who had higher levels of depression and anxiety, who had higher levels of physical and emotional exhaustion, and who had higher levels of active parental internet mediation were more likely to search more health topics online (Table 5).

## Discussion

This study found that two-thirds of adolescents searched health topics online during the COVID-19 pandemic. Prior studies conducted in the United States (Rideout et al., 2018), Serbia (Gazibara et al., 2020), Saudi Arabia (Neumark et al., 2013) and Spain (Jiménez-Pernett et al., 2010) also found that more than half of adolescents reported seeking health information online. In addition, the results found in this study are consistent with prior studies that found adolescents searched a variety of health topics including exercise, fitness, nutrition, and mental and sexual health, with exercise and fitness being the most common health topics searched (Jiménez-Pernett et al., 2010; Neumark et al., 2013; Park and Kwon, 2018; Rideout et al., 2018). These results indicated that online information was an important source for adolescents to obtain health information. However, a study reviewed websites and found that very few webpages were written specifically for adolescents and suggested that governments invest in co-designing excellent-quality and more interactive health information online that better targets an adolescent audience (Ruan et al., 2021).

In addition, the present study found that half of adolescents searched mental health topics online, and that adolescents who had higher levels of depression and anxiety were more likely to search health information and mental health information online. Prior studies also found that individuals with psychological distress were more likely to engage in seeking help online (Pretorius et al., 2019) and in searching for health information (Gallagher and Doherty, 2009; Rowlands et al., 2015). An Australian study found that young women experiencing "stigmatized" conditions were more likely to search health information online (Rowlands et al., 2015). A review study indicated that the benefits for young people who use online help-seeking searches of mental health included anonymity, immediacy, ease of access, inclusivity, shared experiences and a sense of control over the help-seeking journey (Pretorius et al., 2019). These results were consistent with help-seeking models (Rickwood et al., 2005) that showed that when young people had psychological needs and could easily access online mental health information, they were more willing to seek mental health resources online. Studies have established that young people were open to accessing mental health information online, as well as pursuing mental health support online (Oh et al., 2009; Horgan and Sweeney, 2010).

**Table 4.** Factors related to adolescent pursuit of health information online

| | Pursuit of health information online | | | Pursuit of mental health information online | | |
|---|---|---|---|---|---|---|
| | OR | 95% C.I. | *p* value | OR | 95% C.I. | *p* value |
| Gender (boy = 1, girl = 0) | 1.35 | 1.01–1.79 | 0.0395 | 0.90 | 0.69–1.18 | 0.4459 |
| Age | 1.11 | 1.02–1.22 | 0.0207 | 1.28 | 1.18–1.39 | <.0001 |
| Academic performance | 0.93 | 0.79–1.10 | 0.4213 | 0.97 | 0.84–1.14 | 0.7383 |
| Internet learning time | 1.00 | 0.99–1.01 | 0.4833 | 1.00 | 0.99–1.01 | 0.8523 |
| Internet recreation time | 1.00 | 0.99–1.00 | 0.4968 | 1.00 | 1.00–1.01 | 0.5161 |
| Bonding social capital | 0.84 | 0.70–1.01 | 0.0647 | 0.95 | 0.80–1.13 | 0.5600 |
| Competence | 1.03 | 0.84–1.25 | 0.8098 | 1.11 | 0.93–1.33 | 0.2458 |
| Autonomy | 1.09 | 0.92–1.30 | 0.3195 | 1.06 | 0.90–1.24 | 0.5057 |
| Relatedness | 0.93 | 0.77–1.13 | 0.4898 | 1.07 | 0.89–1.28 | 0.4701 |
| eHealth literacy | 2.19 | 1.71–2.79 | <.0001 | 1.38 | 1.11–1.71 | 0.0032 |
| Depression and anxiety | 1.17 | 1.04–1.32 | 0.0111 | 1.34 | 1.20–1.49 | <.0001 |
| Physical exhaustion | 1.13 | 0.91–1.41 | 0.2553 | 1.19 | 0.98–1.45 | 0.0866 |
| Emotional exhaustion | 1.15 | 0.92–1.44 | 0.2089 | 1.67 | 1.35–2.06 | <.0001 |
| Parental restrictive Internet mediation | 1.02 | 0.84–1.22 | 0.8774 | 1.03 | 0.87–1.22 | 0.7663 |
| Parental active internet mediation | 1.22 | 1.01–1.48 | 0.0443 | 1.12 | 0.94–1.33 | 0.2017 |

*Note*: (1) N = 1,236. (2) Multiple logistic regression was conducted. (3) Pursuit of health information online: yes n = 931 no n = 305, pursuit of mental health information online: yes n = 596 no n = 640.

**Table 5.** Factors related to the number of health topics searched online

| | β | 95% C.I. | *p* value |
|---|---|---|---|
| Intercept | −1.94 | −5.35−−2.55 | <0.0001 |
| Gender (boy = 1, girl = 0) | 0.09 | −0.14–0.30 | 0.4795 |
| Age | 0.48 | 0.08–0.22 | <0.0001 |
| Academic performance | −0.11 | −0.23–0.02 | 0.0991 |
| Internet learning time | 0.01 | −0.01–0.01 | 0.7758 |
| Internet recreation time | 0.01 | −0.01–0.01 | 0.9089 |
| Bonding social capital | −0.10 | −0.25–0.03 | 0.1283 |
| Competence | 0.18 | 0.03–0.32 | 0.0214 |
| Autonomy | −0.01 | −0.13–0.14 | 0.9576 |
| Relatedness | 0.00 | −0.14–0.15 | 0.9383 |
| eHealth literacy | 0.52 | 0.34–0.69 | <0.0001 |
| Depression and anxiety | 0.28 | 0.20–0.36 | <0.0001 |
| Physical exhaustion | 0.17 | 0.01–0.33 | 0.0539 |
| Emotional exhaustion | 0.26 | 0.07–0.42 | 0.0052 |
| Parental restrictive internet mediation | 0.00 | −0.11–0.17 | 0.6572 |
| Parental active internet mediation | 0.18 | 0.03–0.32 | 0.0155 |

*Note*: (1) N = 1,236. (2) Multiple regression was conducted.

Moreover, the results of this study were consistent with those of prior studies (James and Harville II, 2016; Wong and Cheung, 2019; Gazibara et al., 2020) that found individuals with better eHealth literacy were more likely to pursue health information online. Another study also found that the internet skill level of adolescents was associated with their pursuit of health information online (Neumark et al., 2013). Results of the present study have shown, however, that adolescents were not as adept at appraising and applying health information as they are at searching for it. Other studies also found that adolescents seldom evaluated search results, had difficulty in selecting appropriate search strings and also had difficulty determining the quality of the information they acquired, which suggests an overall lack of appraisal strategies (Walraven et al., 2009; Jiménez-Pernett et al., 2010; Esmaeilzadeh et al., 2018; Freeman et al., 2018). In addition, in previous studies, adolescents judged their own eHealth literacy much higher than its actual value, and those studies suggested implementing education to strengthen adolescents' eHealth and critical literacy (Maitz et al., 2020; McKinnon et al., 2020). At least one study associated exposure to credible sources of health information online with higher eHealth literacy and suggested that credible health information resources online be incorporated into school health education curricula (Ghaddar et al., 2012). Schools could implement eHealth literacy combined with critical media literacy programs to strengthen adolescents' eHealth literacy competence and enhance their pursuit of health information online.

This study positively associated active parental internet mediation and bonding social capita with adolescent eHealth literacy. These results indicated the crucial roles of parents and significant others in supporting and providing resources to help adolescents solve online problems and enhance their eHealth literacy. Prior studies have also related active parental internet mediation to adolescent eHealth literacy (Chang et al., 2015). In addition, previous studies have positively associated individual social capital with health information self-efficacy, the scope of health information sources and intentions to pursue health information (Kim et al., 2015). Social capital has also been shown to have a positive effect on technological literacy (Yang et al., 2012). These results underscore the importance of strengthening individual social capital to enhance eHealth literacy. Governments could implement parental internet mediation and eHealth literacy training to

improve eHealth and the online pursuit of health information, which would decrease the digital divide and health inequality among children and adolescents.

The results of this study positively associated internet learning time and self-determination factors such as competence and relatedness with higher levels of eHealth literacy among adolescents. Prior studies also positively related self-efficacy to eHealth literacy (Holch and Marwood, 2020; Maitz et al., 2020). These results were consistent with self-determination theory (Deci and Ryan, 2000), which addresses adolescent competence, autonomy and relatedness, and shows these factors to be crucial for developing eHealth literacy and enhancing the pursuit of health information online. Our results showed that boys reported higher levels of eHealth literacy than girls; however, this difference could have been related to their self-determinism, such as when boys perceived higher levels of competence in appraising the information on the internet. In addition, at least one study has applied self-determination theory and found that the pursuit of health information online offers individuals greater autonomy, competence and relatedness compared with face-to-face office visits with a physician (Lee and Lin, 2016). Similarly, another study found that individual competence increases with the use of technology, and learning with peers was more engaging when using digital tools to pursue health information to meet health needs (Scott Duncan et al., 2019). These results suggest that eHealth services design could incorporate a self-determination perspective to promote adolescent eHealth use.

## Limitations

This study had some limitations. First, this was a cross-sectional study, which limits the information that can be used to infer causality. Second, this study analyzed the dataset from the 2020 Taiwan Communication Survey, and some variables such as the psychological distress scale had a limited number of items, which could reduce the reliability. Third, eHealth literacy was measured based on the adolescents' perceptions, and gaps could exist between perceived eHealth literacy and actual capabilities of searching and evaluating health information. Future studies could assess adolescents in an experimental setting to test their capabilities in searching and appraising health information combined with self-report eHealth literacy. Finally, parental internet mediation was measured based on adolescents' reporting, and future studies could conduct a parent–child dyad study to examine parental influences on adolescents' eHealth literacy, the pursuit of health information online, and physical and mental health outcomes. Despite these limitations, the present study adds to the limited amount of literature that addresses adolescents' pursuit of health information online and the roles of self-determination, eHealth literacy, psychological distress and parental internet mediation.

## Conclusions

During the COVID-19 pandemic, adolescents were vulnerable to problems with mental health. The rates of adolescent pursuit of health information / mental health information and health-related topics online during the COVID-19 pandemic were unknown, as were the associations of self-determination, eHealth literacy, psychological distress and parental internet mediation with adolescents' online health and mental health information seeking. Our results showed that two-thirds of adolescents reported searching for

health information online, and about half of adolescents searched mental health information online during the COVID-19 pandemic. The results of this study revealed the significance of the roles of bonding social capital, self-determination and parental active internet mediation in enhancing adolescents' eHealth literacy. Adolescents' levels of competence, eHealth literacy, psychological distress and active parental internet mediation played crucial roles in increasing their pursuit of health and mental health information online. These results implied the need to implement eHealth literacy combined with critical media literacy programs to strengthen adolescents' eHealth literacy competence. Future research could promote a self-determination perspective to develop eHealth literacy intervention, promote the seeking of online health information and underscore the value of appraisal to adolescents.

**Open peer review.** To view the open peer review materials for this article, please visit http://doi.org/10.1017/gmh.2023.44.

**Data availability statement.** The data that support the findings of this study are openly available in the 2020 Taiwan Communication Survey (Phase Two, Year Four): New Communication Technologies & Life Boundary Expansion (D00216) (Data file). Further information is available from the Survey Research Data Archive, Academia Sinica. https://doi.org/10.6141/TW-SRDA-D00216-2 (Chang, 2022).

**Acknowledgements.** Many thanks go to the participant schools and students.

**Author contribution.** Professor Fong-Ching Chang was responsible for the conception and analysis of the study and for the writing of the manuscript. Professor Chingching Chang was the chair to design the Taiwan Communication Survey and the conception of this work. Professor Chen-Chao Tao was the co-chair to conduct the Taiwan Communication Survey and the conception of this work. All authors contributed to manuscript preparation and approved the final manuscript.

**Financial support.** This research received no specific grant from any funding agency, commercial or not-for-profit sectors.

**Competing interest.** The authors declare no conflicts of interest.

**Ethics statement.** Approval of this research was obtained from the Institutional Review Board at Academia Sinica, Taiwan.

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
