## [Reviewer Report]

Dear Editor,

This study is an original research. Each author has contributed significantly to the work. Authors have no conflict of interest regarding the paper. In this study we assessed adolescent pursuit of health information and mental health information online and examined the relationships with eHealth literacy, psychological distress, and parental Internet mediation during the COVID-19 pandemic. We are grateful for your help and kindness.

Best wishes,

Sincerely,

Fong-ching Chang

Professor,

Department of Health Promotion and Health Education

National Taiwan Normal University

---

## [Reviewer Report]

This is an interesting study about psychological factors associated with online health information seeking among adolescents in Taiwan. Although the study does have merit, it has some drawbacks that need to be addressed to make the paper stronger. 

Introduction

"A UK study showed that the time children and adolescents spend online

is increasing" - please specify from what year to what year. Also, provide a reference after this sentence.

Overall, I suggest that the authors omit the location where the studies cited in the Introductions were from, but rather make the first paragraph have more flow to build your rationale for the study. In the present form, the authors keep on citing different studies, but it seems that there is no link between the sentences.

In the second paragraph, I suggest that the authors provide a more detailed rationale on how the COVID-19 pandemic affected people including adolescents' mental health to build your case. 

In the present form, the Introduction is too long, and there is no clear outline of the study rationale. 

The aim is not clearly written. The authors first mention the outcomes of the study and then they mention potential independent predictors of their outcomes. It would be better to number the aims and then articulate which independent predictors might be associated with the study outcomes.

Methods

Please cite the studies which you used to develop your questionnaire. 

The description of the eHEALS reads oddly. The sentences are written under quotation marks, but it is not clear why they stand alone with no link to the latter and former sentences. Please correct. The same applied to the description of other questionnaires.

“A serious of t tests were...” needs to be corrected. What did the authors want to say?

The authors need to describe the rationale for building their regression model and explain the dependent and the independent variables in the manuscript, as there are several of them. The authors need to describe in detail each and every model.

Results

The results are not clear. The authors say in the text that they examined seeking information about mental health online, but in Table 5 they mention the number of topics sought. It is not clear what the outcome is. 

Overall, I suggest removing the regression model for e-health literacy as the results do not materially add much to the main aim of this research.

Please add 95CI for each B in the linear regression.

This manuscript needs to be proofread by a native English speaker.

---

## [Reviewer Report]

Abstract: might profit from some more background (rationale) and details such as exact statistics of amin outcomes. The Impact section does however provide more background and rationale – one part of the Results is a bit hard to process (i.e., ‘had higher levels of active Internet mediation by parents were more likely to have higher levels of eHealth literacy’).

Introduction: authors report relevant background on adolescents’ search for health information world-wide and the difficulties they encounter. The importance of eHealth literacy to arrive at the appropriate eHealth services is also stressed. A definition of Health literacy is provided but maybe the distinction between perceived and performance based eHealth literacy would be helpful for readers as well (as it seems important the way it is phrased here). I wondered at some point what age group is aimed at here; adolescents 10 to 19 but also student findings are presented. Relevance of parental mediation and in terms of lack of evidence on Health literacy in adolescents is clear. Roughly all relevant concepts have been substantiated in the Introduction accept the role of self-determination.

Methods:

What is meant by ‘representative adolescent nationals’: no diversity in terms of ethnicity? Was a range selected in terms of age? How were respondents approached in the schools? More details are needed here – in order to receive more information on sample selection etc.

Regarding instruments used, a self-administered questionnaire was developed based on previous studies. What studies, please refer to the sources (later on this is provided though). Please provide a more information on psychometrics (particularly validity ‘the extent to which an instrument measures what it asserts to measure’) such as for the eHealth literacy questionnaire and Bonding social capital. The psychological distress scale also needs to be better substantiated: 4 items with two on the main depression symptoms and two on anxiety is very limited.

In the statistical analysis plan, please also define predictors and outcomes more clearly. Were the multiple (logistic) regression models corrected for confounders, were covariates age, gender etc. included in the models? No information on confidence intervals nor p-values provided. This part is insufficiently elaborated; also no information on informed consent provided / anonymity etc.

Results:

No age range of participants provided in Table or Text. The difference between boys and girls in eHealth literacy might be related to by their self-determinism (they think they are more competent in appraising the information on the internet). For proper interpretation, please re-phrase sentences like: ‘….that adolescents who were boys, had higher levels of eHealth literacy, had higher levels of depression and anxiety, had higher levels of parental active Internet mediation were more likely to seek health information online’ into ‘adolescents who were boys, who had higher levels of eHealth literacy, who had higher levels of depression and anxiety, or who had higher levels of parental active Internet mediation were more likely to seek health information online.’ The interpretation of the findings is hard to follow otherwise (please also adapt similar phrasings throughout the manuscript). Tables are simple but informative; level of analyses relatively basic. Age would have been valuable to include as a factors as well but middle vs high school may serve as a proxy for age here.

Discussion:

Main findings are presented and reflected on in light of the literature. Important that authors also stress the practical implications such as the need to implement eHealth literacy combined with critical media literacy programs to strengthen adolescents’ eHealth literacy competence since this is actually not measured in this study. Also, the roles of social bonding and parental mediation in eHealth literacy are interesting and of value. Meta-data of adolescents in an experimental setting where they are asked to search for health related information could be combined with self-reports in future research. Self-determination theory is now adequately related to the findings and topic of study (opposed to the lack of it in the Introduction). The Conclusion section seems more a full replication/recap of the results; this could be more balanced. So include a reiteration of the research problem with a summary of only the key findings and a short discussion of the implications of your research also for future research.

---

## [Reviewer Report]

Dear Dr. Bass: 

We appreciate the reviewers’ helpful comments. We have made changes in response to the comments. We are grateful for your help. 

Best wishes,

Sincerely,

Fong-ching Chang